# Longitudinal Changes of Deep and Surface Learning in a Constructivist Pharmacy Curriculum

**DOI:** 10.3390/pharmacy8040200

**Published:** 2020-10-26

**Authors:** Andries S. Koster, Jan D. Vermunt

**Affiliations:** 1Department of Pharmaceutical Sciences, Utrecht University, Universiteitsweg 99, 3584 CG Utrecht, The Netherlands; 2School of Education, Eindhoven University of Technology, Groene Loper, 5612 AE Eindhoven, The Netherlands; J.D.H.M.Vermunt@tue.nl

**Keywords:** learning approaches, deep approach, self-regulated learning, constructivist curriculum, longitudinal changes

## Abstract

In the undergraduate Pharmacy program at the department of Pharmaceutical Sciences, Utrecht University, an educational model is used that is aimed at the development of deep and self-regulating learning. It is, however, unknown whether these objectives are realized. The aim of this study was to assess longitudinal changes in processing and regulation strategies of student learning during their progression in the curriculum, that is explicitly based on constructivist principles. Processing strategies (deep vs. stepwise), regulation strategies (self- vs. external), conceptions of learning and orientations to learning were measured with the Inventory of Learning patterns of Students (ILS). Longitudinal data are reported here for students, of which data are available for year 1/2 and year 4/5 (*n* = 90). The results demonstrate that the use of deep processing (*critical thinking* in particular, effect size = 0.94), stepwise processing (*analyzing* in particular, effect size = 0.55) and concrete processing strategies (effect size = 0.78) increases between the bachelor phase (year 1/2) and the master phase (year 4/5). This change is based on the students having a constructivist view about the nature of learning and is mediated through a relatively large increase in the use of self-regulating strategies (effect size = 0.75). We conclude that this six-year undergraduate Pharmacy program effectively stimulates the development of deep and self-regulated learning strategies in pharmacy students.

## 1. Introduction

Pharmacy education, similar to other health care education programs, ultimately aims for the development of a reflective, self-directed practitioner that is committed to life-long learning, is well embedded in a professional environment, and who can bear independent responsibility for pharmaceutical patient care [1,2]. In order to become a ‘reflective practitioner’, students need to adopt deep, reflective learning approaches [3,4], which requires making a transition from dualistic ways of thinking to contextual relativistic reasoning [5] (reviewed by [6,7,8]).

In higher education, the epistemic cognitive development of students from dualism to contextual relativism typically occurs during the progression of students in undergraduate education [5], with the crucial transition to early contextual relativistic thinking at the advanced bachelor or early master level [9,10,11]. Developmental trajectories for individual students can be very different from each other, resulting in large inter-individual variation [9,10] and recent studies have demonstrated that characteristics of the teaching/learning environment, including the role of teachers, can influence the development of student thinking [12,13,14].

It is often stated that study programs, which are designed on constructivist principles, are effective in stimulating the cognitive development of students from dualism to contextual relativism [3,15] and an early empirical study by Muis and Duffy [12] suggests that this is indeed happening. In recent studies, where conventional, lecture-based, teacher-centered medical curricula were compared with more innovative, problem-based and/or integrated, student-centered curricula, it was demonstrated that deep learning and self-regulation processes are stimulated in the innovative curricula, compared to the conventional curricula [16,17,18]. The crucial idea of a constructivist curriculum is that knowledge is built by the students themselves using a deep, reflective learning process and that the teaching/learning environment accommodates, guides or ‘forces’ students along this pathway [19,20]. An analysis by Alt [21] identified eight essential elements of a social constructivist curriculum (Table 1), which nicely illustrates the importance of deep, reflective learning by the students in a teaching/learning environment where mutual interaction among students and between students and teachers is deemed important (see also [20]). Only a carefully designed teaching/learning environment can accommodate or facilitate deep learning and the cognitive development of students in an optimal way [22]. 

The cognitive development of pharmacy students has been studied in a limited number of cases. An early longitudinal study in the Finnish context [23] confirmed that a surface approach to learning and external regulation of learning by students correlated with having dualistic epistemic beliefs and with the belief that learning equates to intake of knowledge (summarized as reproduction orientation or surface learning). On the other hand, a deep approach to learning and self-regulation of learning correlated with having a constructivist view of the nature of learning (summarized as meaning orientation or deep learning). During the three years of this program, reproduction orientation decreased (effect size = 0.68), but meaning orientation did not increase in a statistically significant way. Follow-up studies in large-scale, lecture-based, teacher-centered courses of the same program demonstrated that adoption of deep or surface approaches to learning was dependent on students’ self-regulatory skills and their perception of the teaching/learning environment [24,25]. Statistically significant changes in deep and surface approaches were seen at the group level (due to the large sample size), but extensive inter-individual differences (between students) and within-student changes (between beginning and end of the course) were underlying these findings. In cross-sectional and longitudinal studies from an Australian university, only minor changes in deep and surface approaches were found during a 4 year undergraduate bachelor program [26,27].

Since the year 2000, a pharmacy curriculum has been developed at Utrecht University that is explicitly based on constructivist principles [28,29]. The educational model used for this curriculum aimed at facilitating the development of deep and self-regulated learning, but at the start of the new program, it was unknown whether this objective would be reached. It was, therefore, decided to investigate the learning approaches of students in a longitudinal way. More specifically, processing and regulation strategies of students and their underlying conceptions of learning were investigated using the Inventory of Learning patterns in Students (ILS) [30,31,32].

## 2. Context: The Undergraduate Curriculum

The undergraduate pharmacy program consists of separate bachelor and master programs, according to the principles of the Bologna agreement; in the Netherlands both programs last 3 years. The bachelor and master programs in Utrecht are designed as a continuum, where similar teaching- and learning principles are used (see the second table in ref. [29]). In the bachelor program, the scientific foundation is formed; in the master program, the focus shifts to the application of scientific principles in patient-oriented care (Figure 1). The transition from university-based education to workplace-based practice is made in the master program and subsequent postgraduate programs ([33], this volume).

The bachelor and master curriculums are based on constructivist principles (see Table 1) and use problem-based and project-based education as basic educational formats. Many courses are designed as integrated, thematic, or methodological courses (see Figure 1). In the integrated courses, scientific content is presented by teachers from at least two different disciplines (e.g., medicinal chemistry, pharmacology, and pharmaceutics in the course *Binding and action of medicines* in year 1). In thematic courses, the presentation of disciplinary content is limited to knowledge that is relevant for the theme only, while the emphasis is on the application of procedures and methods rather than on the science content in methodological courses. Skills are taught and practiced during the whole curriculum, but integrated in the courses (not shown explicitly in the figure). The development of skills is structured along different lines (e.g., pharmaceutical calculations, oral communication, and management) and students’ outcomes are recorded in an electronic portfolio; additional training and practicing for underperforming students is offered in skills labs. Approximately 25–40% of the time allotted to a course is used for student–teacher contact—partially in the form of plenary lectures (up to a maximum of 30 h for a course of 200 h) and partially in the form of small group sessions (5–15 students, for problem-based or project-based work) and lab sessions, practical experiments, or training (up to 30 students per group). The remainder of the course time is available for working on assignments or projects and for individual study. Exams and assessments of skills are integrated within the courses, with opportunities for additional testing outside the planned courses. Every student has a personal tutor, who guides the student in making choices and helps them in reflecting on their skills development and study progress. 

## 3. Methods

For the purpose of this study, data were extracted from a larger dataset, collected in the years 2005 to 2014. Every year (March–May), all students were asked to fill in the 120-item Dutch version of the ILS [30] during class hours; response rates varied between 20 and 74%. Because we wanted to make paired (within-student) comparisons, results of every year for the same students were combined based on student registration numbers. Within the whole dataset, data of 90 students were available for year 1 and/or 2 (bachelor) and year 4 and/or 5 (master). If data were available for the same student from year 1 and year 2, the results were averaged; the same was performed if data from year 4 and year 5 were available. The final dataset contained the results of students who started their study in September 2005, 2006 or 2008. The sample is comparable (in terms of gender composition and study results in the first year of study) to a larger dataset collected in the years 2005 to 2010 (*n* = 491; data not shown).

In the ILS, five processing strategies (ils01 = *relating and structuring*, ils02 = *critical processing*, ils03 = *memorizing and rehearsing*, ils04 = *analyzing*, and ils05 = *concrete processing*), five regulation strategies (ils06 = *self-regulation (learning process and outcomes)*, ils07 = *self-regulation (learning contents)*, ils08 = *external regulation (learning process)*, ils09 = *external regulation (learning outcomes)*, and ils10 = *Lack of regulation*), five conceptions of learning (ils11 = *construction of knowledge*, ils12 = *intake of knowledge*, ils13 = *use of knowledge*, ils14 = *stimulating education*, and ils15 = *cooperative learning*) and five learning orientations (ils16 = *personally interested*, ils17 = *certificate oriented*, ils18 = *self-test oriented*, ils19 = *vocation oriented*, and ils20 = *ambivalent*) are probed [30,31]. All items were scored on 5-point Likert scales (1–5). Scale scores of the ILS were calculated by averaging the scores of the constituent items; Cronbach’s alpha varied between 0.64 and 0.85. In two cases (ils04 in year 1/2 and ils03 in year 4/5), Cronbach’s alpha was lower, 0.57 and 0.40, respectively. 

Because we were mainly interested in how deep and surface learning are linked to regulation of learning and the underlying conceptions of learning, processing and regulation scales were combined to higher-order scales as follows (illustrated in Table 2 and detailed in Appendix A):
ils0102: *Deep processing* was calculated as the average of ils01 (relating and structuring) and ils02 (critical processing);ils0304: *Stepwise processing* was calculated as the average of ils03 (memorizing and rehearsing) and ils04 (analyzing);ils0607: *Self-regulation* was calculated as the average of ils06 (self-regulation of learning process and outcomes) and ils07 (self-regulation of learning contents);ils0809: *External regulation* was calculated as the average of ils08 (external regulation of learning process) and ils09 (external regulation of learning outcomes).

Scores for all ILS scales in the bachelor program (year 1 and/or 2) were compared to scores in the master program (year 4 and/or 5) using Student’s paired *t*-test. Effect sizes (Cohen’s *d*) were calculated as the mean difference between the bachelor and master scores, divided by the standard deviation. Effect sizes were considered small (0.2 to 0.5), medium (0.5 and 0.8) or large (0.8 and larger), according to Cohen [34]. 

Functional relationships between conceptions of learning, regulation strategies and processing strategies were mapped using path analysis. With this technique, it is possible to estimate a network of interdependent relationships, based on the correlation matrix of the measured variables; the best-fitting model is selected on statistical arguments. SPSS (version 25; IBM corporation, Armonk, USA) was used for basic statistical analysis (*t*-test, correlations); Systat-13 (Evanston, IL, USA) was used for path analysis, using the RAMONA routine.

## 4. Results

From the data (Table 3), it is clear that important changes occur in the learning approach of students when they move from the bachelor program (year 1 and/or 2) to the master program (year 4 and/or 5). In contrast to cross-sectional studies, relatively small intra-individual changes reach statistical significance due to the paired setup of the design of this study. In order to obtain a large enough sample, data of year 1 had to be combined with data of year 2, and data of year 4 were combined with data of year 5. This means that the observed changes can be two (*n* = 36), three (*n* = 47) or four (*n* = 7) years apart from each other for an individual student. Nevertheless, the results give a clear picture about the medium- to long-term changes in student learning.

A statistically significant increase in deep approach to learning, expressed as an increase in *deep processing* (ils0102, effect size = 0.55) and *self-regulation* (ils0607, effect size = 0.75) is clearly observed. When analyzed on a deeper level, the change in deep processing appears to be exclusively a result of a change in *critical processing* (ils02, large effect size); no change is seen in the ils01 scale *relating and structuring*. In contrast, relatively minor changes are seen in the surface approach to learning, as expressed by a small increase in *stepwise processing* (ils0304), which is caused by an increase in *analyzing* (ils04) rather than an increase in *memorizing and rehearsing* (ils03).

Both deep processing and stepwise processing increase between the bachelor and master programs. The balance between the deep and stepwise learning process does not change when students move from the bachelor to the master program. Self-regulation with respect to the learning contents increases strongly (effect size = 0.92), but self-regulation with respect to the process of learning changes less (effect size = 0.31). This means that a change in supposed metacognitive activities of the students is mainly concerned with the outcome of their learning, but that relatively minor changes in reflection on the learning process itself seem to occur in this stage of their study.

The variability between students in processing and regulation strategies is extensive, as can be seen already from the large standard deviations (Table 3). This phenomenon is further illustrated by the results of eleven individual students, of which data of four study years were available (Figure 2). Within-student changes between year 1, 2, 4 and 5 are present but are minor compared to the inter-student variability. It can also be seen that year-to-year changes do occur, but that the position of an individual student (indicated by a unique color in the figure), when compared to other students, is remarkably stable. Finally, it is suggested that changes in deep processing are mainly occurring between year 4 and year 5, while changes in self-regulation are occurring between year 2 and year 4, i.e., in the second half of the bachelor program.

In the first two years, pharmacy students appear to have mixed conceptions of learning. The conceptions that learning consists of *intake of knowledge* (ils12), *use of knowledge* (ils13) or *construction of knowledge* (ils11) have more or less equal weight in the beginning of the bachelor program and do not change considerably (see Table 3). A relatively small increase in the idea that learning consists of construction of knowledge is seen (ils11, effect size = 0.39). Remarkably, at the same time, it can be observed that the idea of learning being use of knowledge decreases (ils13, effect size = 0.38) between the beginning of the bachelor and the beginning of the master programs. This may be related to the science-based nature of the bachelor program. Students are becoming less vocation oriented (ils19) concomitantly. An increase in the self-test-oriented learning orientation (ils18) is consistent with the increase in a deep learning approach (i.e., deep processing and self-regulation).

The relationships between the three main learning conceptions and regulation and processing strategies were further investigated using path analysis of the correlations between scale scores (Appendix B). For the bachelor phase, the best-fitting model established clear relationships between a constructivist learning conception, self-regulation, and deep processing (Figure 3, left panel). This means that students who score higher on the scale of *construction of knowledge* (ils11) tend to also score higher on the scale of *self-regulation* (ils0607) and the scale of *deep processing* (ils0102). Forty percent of the variance in the self-regulation score and 29% of the variability in the deep processing score can be explained in terms of the underlying, in this case constructivist, conception of learning. In contrast, 32% of the variance in *stepwise processing* (ils0304) can be explained in terms of a combination of *self-regulation* (ils0607) and *external regulation* (ils0809). External regulation of student learning is dependent on the students having a dual conception of learning, where both *construction of knowledge* (ils11) and *intake of knowledge* (ils12) contribute; together, these two variables explain 16% of the variance in the external regulation score.

When students move from the bachelor program to the master program, two subtle changes in the relationships between their conceptions of learning, regulation strategies and processing strategies occur (Figure 3, right panel). External regulation is no longer dependent on having *intake of knowledge* (ils12) as a conception of learning, but is now influenced by having *use of knowledge* (ils13) as a learning conception (19% of the variance explained). The variance in deep processing strategy is no longer explained in terms of *self-regulation* (ils0607) only, but is also influenced by *external regulation* (ils0809). In view of the observation that *external regulation of the learning process* (ils08) increases between the bachelor and master phases, while *external regulation of learning outcomes* (ils09) decreases (see Table 3), this probably means that characteristics of the teaching/learning environment (e.g., teacher behavior or assignments) have an increasing impact on the learning process of students, while students become more and more dependent on self-regulation as far as the content of learning is concerned. The deep processing learning strategy and part of the stepwise processing learning strategy during the master phase remain dependent on self-regulation and on having a constructivist view of the nature of learning. The strength of the influence of self-regulation on stepwise processing increases in the master program (path coefficient = 0.78), compared to the bachelor program (path coefficient = 0.24).

## 5. Discussion

In this study, it has been possible to demonstrate statistically significant longitudinal changes in the processing and regulation of many learning strategies in addition to changes in the relationships between processing strategies, regulation strategies and the underlying conceptions of learning. In earlier studies, it has been difficult to unambiguously demonstrate increases in the deep learning approaches of students in higher education, which are theoretically supposed to occur [35]. This may be caused by earlier studies being performed in conventional, teacher-centered curricula [17,23,24], but also by large inter- and intra-individual differences [24,25]. Comparable large inter-individual variations were found in this study, expressed as large standard deviations, and paired comparisons between the bachelor and master phases could be made due to the availability of longitudinal data. It was shown that changes in learning approaches, with medium to large effect sizes, can be demonstrated. The conclusions may be weakened somewhat because of a highly variable response rate between years and between courses, but the paired measurement is not affected by this (as a cross-sectional, non-paired measurement would have been). Two further limitations are the sample size (small compared to the total number of students) and a relatively low reliability of 2 out of 40 measurements scales (see Methods). We estimate that these limitations will not be so large as to compromise our results and conclusions.

From the results of this study, it can be concluded that constructivist opinions and self-regulation have an overall importance for explaining variability in learning processes both in the bachelor and the master phase of this curriculum. Already in the bachelor phase, student views about the nature of learning as *construction of knowledge* are reasonably high (3.25 ± 0.46, mean ± s.d.), in addition to the opinion of learning being *intake of knowledge* (3.72 ± 0.43) or *use of knowledge* (3.68 ± 0.45). A small, but statistically significant increase (effect size = 0.39) in the constructivist opinion is seen two to four years later, when students progress to the master phase of the program. Self-regulation of learning is strongly dependent on having a constructivist opinion: 40% (the bachelor phase) or more (the master phase) of the variance in self-regulation is explained by the constructivist conception of learning. Both deep and stepwise processing of learning, in turn, are influenced by self-regulation characteristics of the students. The increase in self-regulation between the bachelor and master phases, mostly with respect to the content of learning, explains important changes in deep processing strategies (effect size = 0.55), mainly as a change in *critical processing* (effect size = 0.94).

The overall conclusion must be that pharmacy students in this curriculum use deep learning approaches (deep processing and self-regulation, based on constructivist opinions) already in the bachelor phase and that these deep learning approaches are further enhanced between the bachelor and the master phase. In this respect, the results compare favorably with earlier studies from the Australian and Finnish contexts. In a 4 year program at the University of Sydney, measures of the meaning-directed approach decreased during the first half of the curriculum, while only a ‘recovery’ was seen during the second half of the curriculum [26,27]. Measures of meaning-directed orientation, deep approach, self-regulation and a constructivist conception of learning did not change in a statistically significant way in a 3 year undergraduate program at the University of Helsinki [23]. In longitudinal studies of medical curricula in California (USA) and Edinburgh (UK), no statistically significant changes in deep approach were seen between year 1 and year 4 or 5, but this was interpreted as students already having a relatively deep approach at the beginning of the curriculum due to the constructivist nature of those curricula [36,37]. We cannot be certain that the absence of changes in the Finnish and Australian contexts were due to the conventional, lecture-based, teacher-centered, character of the curricula, but it should be mentioned that extensive curriculum change processes are being effected in Helsinki in more recent years [3,38].

A second result of this study is that surface learning, expressed as stepwise processing and external regulation, also appears to be influenced by the constructivist conception of learning. In both the bachelor and the master phases, stepwise processing is dependent on seeing learning as construction of knowledge. This is effected indirectly via two pathways and mediated by the effect of a constructivist view of external regulation and the effect of self-regulation on stepwise processing. Changes in surface learning between the bachelor and master phases are relatively minor and consist of a small increase in stepwise processing (effect size = 0.30, caused by a change in the analyzing score), a decrease in external regulation of learning outcomes (effect size = 0.60) and an increase in external regulation of the learning process (effect size = 0.50). This could be interpreted as a result of the way the curriculum is organized, where attention for the process of learning in the form of instructions, tasks, assignments, and teacher roles can be perceived by the students as external regulation of the learning process.

A last result of this study is that deep processing in the master phase becomes partly dependent on external regulation (path coefficient = 0.25) in addition to the dependency on self-regulation (path coefficient = 0.49). This can also be interpreted as a result of the curriculum, where assignments are explicitly formulated in such a way that deep processing, consisting of relating and structuring and critical processing, is stimulated. The way in which problem-based and project-based education is organized could be perceived as external regulation by the students.

The results of this study confirm that the conceptual understanding of students about the nature of learning affects their learning strategies, and that this relationship is mediated via the regulation of learning. Having a constructivist understanding (i.e., seeing learning as knowledge construction) appears to be particularly important for self-regulation of learning, as this influences all aspects of deep learning (relating and structuring and critical processing) and stepwise learning (memorizing and rehearsing and analyzing) investigated in this study. Constructivist opinions have a secondary effect on stepwise learning, which is mediated via external regulation of learning, together with the contribution of other conceptions of learning (intake of knowledge during the bachelor phase or use of knowledge during the master phase). These relationships between conceptions of learning, regulation strategies and processing strategies are consistent with observations in post-secondary education in Hong Kong [39] and advanced psychology students in the Netherlands [40]. The functional relationships between conceptions of learning, regulation strategies and processing strategies have been synthesized in an integrated model of student learning [30], which has subsequently been found useful in numerous contexts (reviewed by Vermunt and Donche [31]). It is now confirmed that in the context of pharmacy education, essentially the same relationships between conceptual understanding of learning, regulation of learning and processing of learning exist, and that substantial changes in deep learning (characterized by self-regulation and deep processing) can occur when students advance from the bachelor phase to the master phase.

In the investigated curriculum, students develop a higher degree of self-regulation (effect size = 0.75), in particular with respect to the content of learning, and they perceive higher degrees of critical processing (effect size = 0.94; an aspect of deep processing), analyzing (effect size = 0.55; an aspect of stepwise processing) and concrete processing (effect size = 0.78) during the master phase, compared to the bachelor phase 2–4 years before. We do not know whether students are developing this increased self-regulation and change in learning processes as a result of characteristics of the teaching/learning environment or whether the teaching/learning environment only facilitates or accommodates an endogenous cognitive development process. It is highly likely that the cognitive development of students from dualistic thinking to contextual relativism is ongoing [7,8,10,11] and that this development is strengthened by characteristics of the teaching/learning environment. In particular, the integration of course content and the use of problem- and project-based learning can be important [18,41]. Integrative teaching of content in student-activating educational formats such as problem-based and project-based learning can be effective in stimulating deep learning by creating cognitive frictions [42]. Student–student interaction, while working on projects, and student–teacher interactions, during problem-based learning sessions and in individual tutorials, can contribute to developing multiple perspectives and cooperative dialogues (compare Table 1). These curricular aspects appear to more effective in stimulating deep, reflective learning when introduced in a curriculum-wide—systematic—way [16,41], where contextual factors and student characteristics are taken into account [22]. The conclusion from our study can be that several design elements of the curriculum contribute to the observed changes in self-regulation and deep and stepwise learning processes.

The importance of designing a teaching/learning environment, that accommodates and strengthens the appropriate cognitive development of students can be illustrated as follows (Figure 4). The scheme assumes that the cognitive development of students from dualistic to contextual relativistic thinking is endogenous and can be strengthened by the way in which the teaching/learning environment is constructed and/or implemented [29]. In a constructivist curriculum, the cognitive development of students, expressed as deep, self-regulated and reflective learning, is fostered by characteristics of the teaching/learning environment. The teaching/learning environment can be more (or less) stimulating for this development depending on whether the content is delivered in an integrated or non-integrated way, the choices for educational formats and the way student–student and student–teacher interactions are organized. Whether the full potential of a curriculum is achieved will also depend on many contextual and student characteristics [22], in particular their critical reflection and metacognitive capabilities.

## 6. Conclusions

The result of this study demonstrates that the use of deep processing (in particular *critical thinking*), stepwise processing (in particular *analyzing*) and concrete processing learning strategies increases in the undergraduate pharmacy curriculum of Utrecht University between the bachelor phase (year 1/2) and the master phase (year 4/5). This change is based on an increase in constructivist views about the nature of learning and mediated through a relatively large increase in the use of self-regulating strategies. These changes are most likely a reflection of endogenous cognitive development, which is strengthened by characteristics of the teaching/learning environment that is explicitly based on constructivist principles.

## Figures and Tables

**Figure 1 pharmacy-08-00200-f001:**
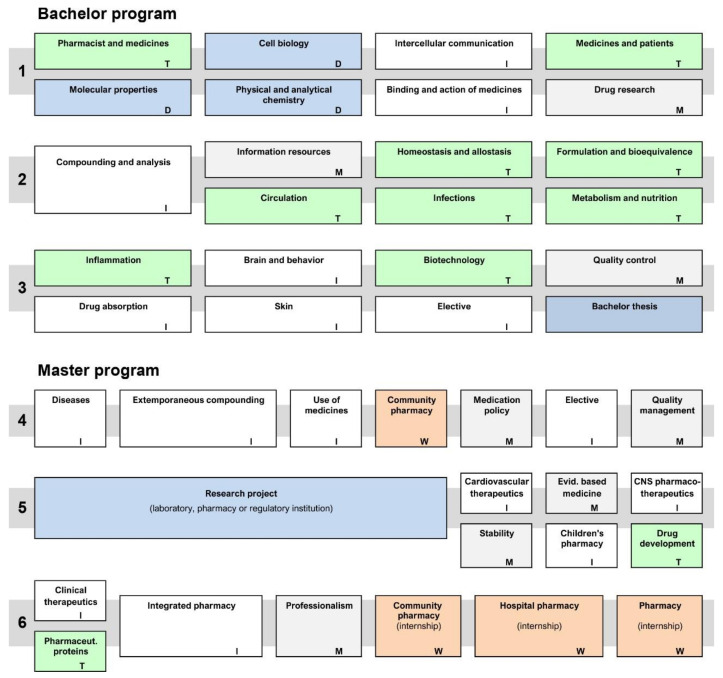
Schematic representation of the bachelor and master curriculums, years 1–6. The curriculum consists mainly of courses, where content of different disciplines is integrated (I, white color) or where relevant themes are used as organizing principles (T, green color); only a few courses in year 1 are monodisciplinary (D, blue color). Other courses emphasize methodological aspects (procedures) rather than content (M, grey color). Internships in community and hospital pharmacy (W, salmon color) are scheduled in the master program.

**Figure 2 pharmacy-08-00200-f002:**
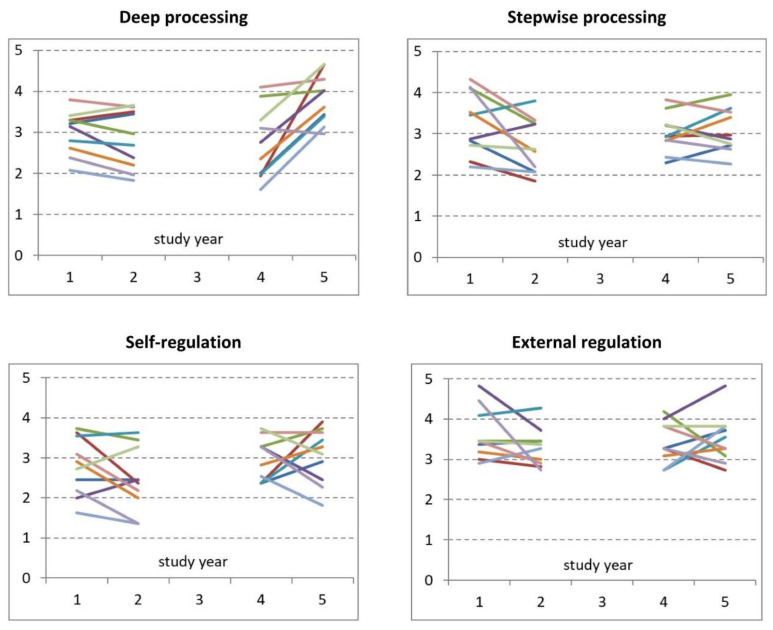
Differences between individual students in deep and surface learning. Scale scores (Y axis) for processing and regulation strategies in year 1, 2, 4 and 5 of eleven individual students are plotted in the figure. Each unique color represents the data of one-and-the-same student.

**Figure 3 pharmacy-08-00200-f003:**
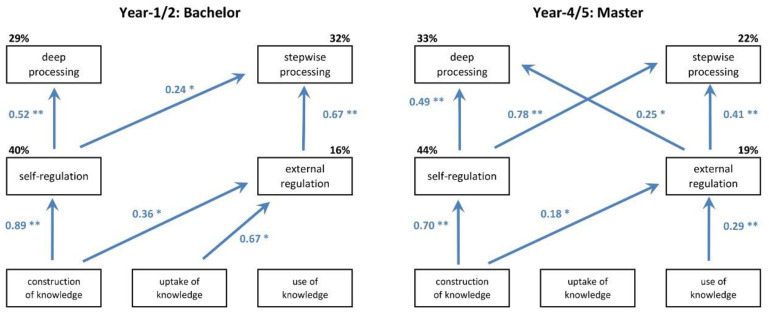
Relations between conceptions of learning, regulation strategies and processing strategies in the bachelor and master programs. Statistically significant positive path coefficients (* *p* < 0.05 or ** *p* < 0.01) are represented by the blue arrows; non-significant paths were excluded. The percentage of explained variances for each for the processing and regulation strategies is indicated on top of the relevant boxes. The fit of both path models was good (RMSEA < 0.05).

**Figure 4 pharmacy-08-00200-f004:**
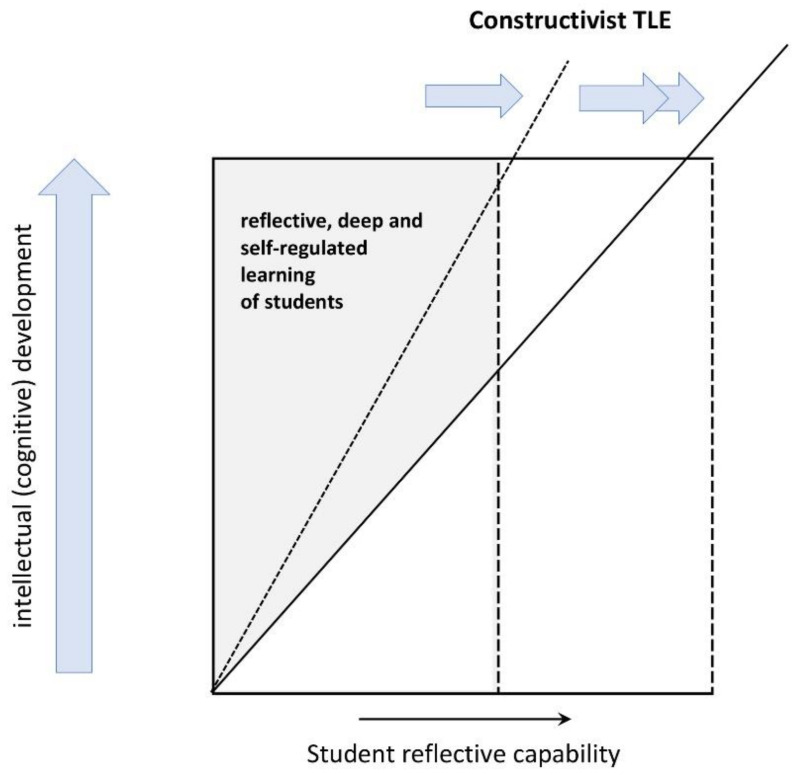
A model for stimulating cognitive development in a constructivist teaching/learning environment (TLE). The longitudinal cognitive development of students (symbolized by the left-hand arrow) is supposed to increase in a curriculum (bottom to top). In a constructivist curriculum, the cognitive development of students, expressed as reflective, deep, and self-regulated learning, is fostered by characteristics of the TLE, which can stimulate cognitive development to a small or larger extent, as symbolized by the slope of the diagonal line (one or two arrows, respectively). Whether the full potential of a curriculum is achieved also depends on the ‘reflective capability’ of the student involved.

**Table 1 pharmacy-08-00200-t001:** Characteristic elements of a constructivist curriculum (after Alt [21]).

1. *Knowledge construction*. This element describes multiple opportunities given to students to investigate real problems, raise questions and search for possible explanations while using various methodological approaches.
2. *In-depth learning.* This category pertains to the extent to which students are given opportunities to deeply explore a certain subject matter, rather than engaging them in surface learning.
3. *Authenticity.* This element deals with giving relevant meaning to the learned concepts and addressing real-life and interesting events, which are related to the studied topic.
4. *Multiple perspectives.* This category refers to presenting complex ideas from several points of view.
5. *Prior knowledge.* This element primarily deals with connecting the subject materials in individual courses to other course topics.
6. *Teacher-student interaction.* This element refers to the teacher role, which includes guidance toward reflection on learning processes.
7. *Cooperative dialogue.* This element refers to dialogical activities during the lesson, in which students can express opinions and original ideas.
8. *Social interaction.* This element includes a variety of learning activities with other students, not necessarily during a lesson.

**Table 2 pharmacy-08-00200-t002:** Characteristics of deep and surface learning.

	Processing Strategies	Regulation Strategies
Deep learning	**Deep processing** scale 01: Relating and structuring scale 02: Critical processing	**Self-regulation** scale 06: Learning process and outcomes scale 07: Learning contents
Item examples	*I try to combine subjects that are dealt with separately in the course into one whole* *I compare my view of a course topic with the views of textbook authors*	*To test my learning, I try to answer questions about the subject matter which I make up myself* *In addition to the syllabus, I study other related literature*
Surface learning	**Stepwise processing** scale 03: Memorizing and rehearsing scale 04: Analyzing	**External regulation** scale 08: Learning process scale 09: Learning outcomes
Item examples	*I memorize lists of characteristics of a phenomenon* *I analyze the separate components of a theory step by step*	*I study according to the instructions in the course manual* *I test my learning solely by completing the questions, tasks, and self-tests in the course material*

In the ILS, each scale is represented by 4 to 8 items; representative Item examples (*in italics*) of the main processing and regulation scales are taken from the questionnaire, used in this study.

**Table 3 pharmacy-08-00200-t003:** Processing strategies, regulation strategies, conceptions of learning and learning orientations of students in the bachelor and master programs.

ILS Scale		Bachelor (year 1/2)	Master (year 4/5)	*p* (Paired *t*-Test)	Effect Size	
**Processing Strategies**						
ils0102	Deep processing	2.72 ± 0.52	3.05 ± 0.68	< 0.001 ***	0.55	↑↑
ils01	Relating and structuring	3.17 ± 0.60	3.18 ± 0.77	0.86	0.01	
ils02	Critical processing	2.27 ± 0.58	2.92 ± 0.80	< 0.001 ***	0.94	↑↑↑
ils0304	Stepwise processing	2.82 ± 0.57	2.98 ± 0.49	< 0.014 *	0.30	↑
ils03	Memorizing and rehearsing	2.90 ± 0.81	2.91 ± 0.75	0.91	0.01	
ils04	Analyzing	2.74 ± 0.51	3.04 ± 0.58	< 0.001 ***	0.55	↑↑
ils05	Concrete processing	2.62 ± 0.63	3.12 ± 0.65	< 0.001 ***	0.78	↑↑
**Regulation Strategies**						
ils0607	Self-regulation	2.49 ± 0.58	2.92 ± 0.56	< 0.001 ***	0.75	↑↑
ils06	Learning process	2.57 ± 0.65	2.78 ± 0.72	0.015 *	0.31	↑
ils07	Contents	2.31 ± 0.66	2.96 ± 0.75	< 0.001 ***	0.92	↑↑↑
ils0809	External regulation	3.24 ± 0.50	3.30 ± 0.48	0.372	0.12	
ils08	Learning process	3.00 ± 0.53	3.29 ± 0.62	< 0.001 ***	0.50	↑↑
ils09	Learning outcomes	3.49 ± 0.65	3.00 ± 0.98	< 0.001 ***	0.60	↓↓
ils10	Lack of regulation	2.35 ± 0.66	2.97 ± 0.79	< 0.001 ***	0.86	↑↑↑
**Conceptions of Learning**						
ils11	Construction of knowledge	3.25 ± 0.46	3.45 ± 0.57	0.005 **	0.39	↑
ils12	Intake of knowledge	3.72 ± 0.43	3.58 ± 0.63	0.07	0.26	
ils13	Use of knowledge	3.68 ± 0.45	3.44 ± 0.81	0.017 *	0.38	↓
ils14	Stimulating education (teacher)	3.08 ± 0.57	3.15 ± 0.57	0.32	0.12	
ils15	Cooperative learning (students)	3.13 ± 0.61	3.07 ± 0.64	0.52	0.10	
**Learning Orientations**						
ils16	Personally interested	3.07 ± 0.46	3.17 ± 0.61	0.20	0.19	
ils17	Certificate oriented	3.30 ± 0.56	3.19 ± 0.85	0.22	0.16	
ils18	Self-test oriented	3.06 ± 0.81	3.49 ± 1.04	< 0.001 ***	0.46	↑
ils19	Vocation oriented	3.95 ± 0.50	3.03 ± 1.32	< 0.001 ***	1.01	↓↓↓
ils20	Ambivalent	2.02 ± 0.68	2.08 ± 0.92	0.58	0.08	

Data are based on a sample of 90 students who started their bachelor of pharmacy program in September 2005, 2006 or 2008. Statistical significance of the comparison between the bachelor and master programs (Student’s paired *t*-test) is indicated as * *p* < 0.05, ** *p* < 0.01 or *** *p* < 0.001. Effect size [34] is considered small (>0.2; one arrow), medium (>0.5; two arrows) or large (>0.8; three arrows).

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
