# Peer review of "Longitudinal Changes of Deep and Surface Learning in a Constructivist Pharmacy Curriculum"

_pharmacy, 2020, doi:10.3390/pharmacy8040200_

Round 1
Reviewer 1 Report
I am glad to have the opportunity for reviewing these manuscript. I find it very interesting and I think you did a great job by presenting quite novel pharmacy program and showed how it stimulates the development of deep and self-regulated learning strategies in pharmacy students. I include some minor suggestions that I think could contribute to the improvement of your manuscript.
line 87- you refer to the principles of Bologna agreement when mentioning that undergraduate program consists of 3 plus 3 years of study. Bologna process usually implies 3 plus 2 years. Please, reconsider these.
line 124- response rate was 20-74%. This is a big range and limits the results, especially the lower rate. Could you please give the reasons for such a response rate and comment on limitations of this.
line 142- Cronbach alpha values get as low as 0.4 and 0.55- what about this lower reliability coefficients? Are all the scales reliable?
There is small error in the text below table 3. (was is...).
Author Response
I am glad to have the opportunity for reviewing these manuscript. I find it very interesting and I think you did a great job by presenting quite novel pharmacy program and showed how it stimulates the development of deep and self-regulated learning strategies in pharmacy students. I include some minor suggestions that I think could contribute to the improvement of your manuscript.
Authors response: Thank you for your comments and suggestions.
line 87- you refer to the principles of Bologna agreement when mentioning that undergraduate program consists of 3 plus 3 years of study. Bologna process usually implies 3 plus 2 years. Please, reconsider these.
Authors response: The 3+3 year program is indeed somewhat unusual in the Bologna context. The text has been changed to indicate that having a separate bachelor and master program is in accord with Bologna, but that the 3+3 year program is specific for the Netherlands.
line 124- response rate was 20-74%. This is a big range and limits the results, especially the lower rate. Could you please give the reasons for such a response rate and comment on limitations of this.
Authors response: The logistics of this longitudinal study are complicated. As a consequence the response rate varied from year to year and from course to course (questionnaire were collected during courses by the course coordinators). However, for this study paired data were used of those students, of which data from year 1/2 and year 4/5 were available. This comparison is not compromised by response rates, as a cross-sectional, non-paired study would have. A short discussion has been added to the Discussion section.
line 142- Cronbach alpha values get as low as 0.4 and 0.55- what about this lower reliability coefficients? Are all the scales reliable?
Authors answer: Only two scales were in this low range, unfortunately. The text has been modified to indicate this.
There is small error in the text below table 3. (was is...).
Corrected.
Reviewer 2 Report
Is it possible to include the questionnaire (translated in English) as a supplement to your manuscript?
I would like to see the title for y-axis in the Figure 2
Also, for Figure 1 the letter D is missing from two of the blue boxes.
These are minor flaws, otherwise I am very pleased with the manuscript: it adds important base of knowledge concerning the deep learning and understanding in science-based curriculum. Also, as there is scarce literature about pharmacy curriculum and teaching, this is an important paper.
Author Response
Is it possible to include the questionnaire (translated in English) as a supplement to your manuscript?
Authors answer: The complete questionnaire is quite long and can be obtained free of charge from the second author, who prefers to be in contact with users of the ILS. For the reader's interest we provided a detailed description of the ILS-scales, each with an example item. Other examples can be found in Table 2.
I would like to see the title for y-axis in the Figure 2
Authors answer: The legend of the figure has been adapted to indicate the scale of the Y-axes.
Also, for Figure 1 the letter D is missing from two of the blue boxes.
Strictly speaking the D was only relevant for the courses in year-1. The bachelor thesis and research project usually are disciplinary, but other possibilities do exist. Therefore, the D has not been used in these curricular elements. The legend of figure 1 has been adapted.
These are minor flaws, otherwise I am very pleased with the manuscript: it adds important base of knowledge concerning the deep learning and understanding in science-based curriculum. Also, as there is scarce literature about pharmacy curriculum and teaching, this is an important paper.
Authors answer: Thank you for this comment. We fully agree, but you can imagine that it took some effort to collect these data.